# Building of an Internal Transcribed Spacer (ITS) Gene Dataset to Support the Italian Health Service in Mushroom Identification

**DOI:** 10.3390/foods10061193

**Published:** 2021-05-25

**Authors:** Alice Giusti, Enrica Ricci, Laura Gasperetti, Marta Galgani, Luca Polidori, Francesco Verdigi, Roberto Narducci, Andrea Armani

**Affiliations:** 1FishLab, Department of Veterinary Sciences, University of Pisa, Viale delle Piagge 2, 56124 Pisa, Italy; marta.galgani@gmail.com (M.G.); andrea.armani@unipi.it (A.A.); 2Experimental Zooprophylactic Institute of Lazio and Tuscany M. Aleandri, UOT Toscana Nord, SS Abetone e Brennero 4, 56124 Pisa, Italy; enrica.ricci@izslt.it (E.R.); laura.gasperetti@izslt.it (L.G.); 3Tuscany Mycological Groups Association, via Turi, 8 Santa Croce sull’Arno, 56124 Pisa, Italy; luca.polidori@tiscali.it (L.P.); narducci1956@libero.it (R.N.); 4North West Tuscany LHA (Mycological Inspectorate), via A. Cocchi, 7/9, 56124 Pisa, Italy; francesco.verdigi@uslnordovest.toscana.it

**Keywords:** mushrooms, poisoning, species identification, internal transcribed spacer: genetic dataset, official control

## Abstract

This study aims at building an ITS gene dataset to support the Italian Health Service in mushroom identification. The target species were selected among those mostly involved in regional (Tuscany) poisoning cases. For each target species, all the ITS sequences already deposited in GenBank and BOLD databases were retrieved and accurately assessed for quality and reliability by a systematic filtering process. Wild specimens of target species were also collected to produce reference ITS sequences. These were used partly to set up and partly to validate the dataset by BLAST analysis. Overall, 7270 sequences were found in the two databases. After filtering, 1293 sequences (17.8%) were discarded, with a final retrieval of 5977 sequences. Ninety-seven ITS reference sequences were obtained from 76 collected mushroom specimens: 15 of them, obtained from 10 species with no sequences available after the filtering, were used to build the dataset, with a final taxonomic coverage of 96.7%. The other 82 sequences (66 species) were used for the dataset validation. In most of the cases (n = 71; 86.6%) they matched with identity values ≥ 97–100% with the corresponding species. The dataset was able to identify the species involved in regional poisoning incidents. As some of these species are also involved in poisonings at the national level, the dataset may be used for supporting the National Health Service throughout the Italian territory. Moreover, it can support the official control activities aimed at detecting frauds in commercial mushroom-based products and safeguarding consumers.

## 1. Introduction

Fungi represent a clade of eukaryotic heterotrophic organisms with a vast ecological and economic impact and whose diversity is likely to be in the millions of species [1,2]. Some of them, belonging to the macro-fungi group, can produce mushrooms, distinctive fruiting bodies from an underground mycelium [3], which have always been consumed worldwide. Wild mushrooms are collected in developing countries and rural communities to contribute to their diet or for local financial benefits [4,5,6]; on the other hand, a small group of mushroom species is cultivated on an industrial scale and marketed at global level. Mushrooms are increasingly appreciated since they fully fit in the current food trends addressing health and sustainable choices. They have, in fact, optimum nutritional values, they represent excellent alternatives for vegetarian-style meals, [7] and, not least, they are consistent with a sustainable food supply [8].

Their global market was valued at USD 54,610 million in 2020 and it is expected to reach USD 72,080 million by the end of 2026 [9].

In Italy, where mushroom farming is still unable to satisfy the high national request, cultivated mushrooms are highly imported from China and Eastern Europe. However, there is a long recreational tradition of collecting wild mushrooms, which are more appreciated by Italian consumers with respect to the cultivated ones. This collection takes places in specific months of the year. Wild mushrooms consumed in Italy are mainly “Porcini” (*Boletus* spp.), Chantarelle (*Cantharellus cibarius*), and the nationally called “Ovulo” (*Amanita caesarea*), together with many other species, differently appreciated among regions [10].

The consumption of mushrooms is not exempt from risks. Mushroom poisoning is in fact a significant form of toxin-induced disease that can affect gastrointestinal, neurological, renal, and hepatic systems [11,12]. In the presence of particularly poisonous species, the liver suffers irreparable damage that can be fatal [13]. Additionally, some edible species can become toxic if improperly collected, transported, stored, and cooked [12,14,15]. While there are well-known species, such as *Amanita phalloides*, causing life-threatening poisonings, there is in fact also accumulating evidence of poisonings related to species that have been considered edible and are traditionally consumed [15].

Globally, thousands of poisonings are reported each year, and 80% of them involve unidentified mushroom species [16]. In Italy, mushroom consumption has caused a growing number of poisoning cases, to the point that this intoxication is considered a public health concern [17]. In the period 1998 to 2017, the Poison Control Centre of Milan, the national reference center for mushroom poisoning, received 15,864 emergency calls. In the same period 46 deaths were recorded, even though a greater number of fatal cases is estimated [18]. In most cases, poisoning occurs because of species misidentification by amateur mushroom hunters [13], as some poisonous species resemble the edible ones in color, size, and general shape.

In Italy, the Law of 23 August 1993 No. 352 [19] and the Presidential Decree of 14 July 1995 No. 376 (DPR 376/1995) [20] establish the mushroom market rules and list the wild or cultivated species that can be placed on the market in the fresh state. However, regions may also integrate this list by inserting other species of local interest. Imported species can be placed on the market if certificated as edible by the competent authorities of the country of origin and by the Italian Local Health Authorities (LHA). Currently, the Italian national list comprises around 150 mushroom species. Importantly, before being allowed to sell wild mushrooms, local mushroom dealers must pass an exam in wild mushroom identification and knowledge of any special treatment required by certain species before consumption [10].

The DPR 376/1995 [20] also set up the mycological inspectorates, composed of a team of expert mycologists dealing with the official control of mushrooms and providing a free recognition service for mushrooms picked by private citizens, which is fundamental to prevent poisoning incidents. Additionally, they are consulted for identifying mushroom species in case of poisoning. The identification by expert mycologists is based on the observation of phenotypic characters, macroscopic structures of the fruit body, or microscopic structure of the spores [2]. However, many poisonous mushrooms resemble edible ones and all genera that contain poisonous mushrooms also include many non-poisonous and edible mushrooms [16]. Additionally, phenotypic characters are strongly affected by environmental conditions, hybridization process, cryptic speciation, and convergent evolution [21,22,23]. Therefore, the phenotypic approach may not always perform well, and it is particularly challenging for processed products (dried, canned, or complex multispecies matrices) or clinical samples (such as vomit or feces) [24,25]. Alternatively, the microscopic inspection of tissues and spores and the chemical composition evaluation are also largely used for the species identification of mushrooms. Technological advancement in the field has led to the increasing use of sophisticated techniques for identifying mushrooms, such as gas chromatography–ion trap mass spectrometry of mushroom metabolites [26] or protein profiling based on matrix-assisted laser desorption/ionization mass spectrometry (MALDI-TOF MS) [27], among others.

However, DNA analysis complementing phenotypic observations is the method of choice for specific identification of mushroom species, thanks to its rapidity and relatively low cost [21,28]. Current methods are almost entirely based on analyses of PCR-amplified nuclear rRNA genes, particularly the Internal Transcribed Spacer (ITS) region [2], which has been chosen as a universal DNA barcode marker for fungi due to its high inter-species variation [29]. The ITS region (~600–800 bp) contains two non-coding regions (ITS-1 and ITS-2) separated by the highly conserved small subunit 5.8S rRNA gene [30]. In general, this marker is sequenced and compared to public reference DNA libraries, that are imperative tools for taxonomic assignments. Crucial to this approach is a comprehensive and validated reference set of DNA sequence data from target species to provide accurate and verifiable molecular identification [31]. Several fungi ITS sequences are available in the NIH genetic sequence database, GenBank (https://www.ncbi.nlm.nih.gov/genbank/, accessed date 24 May 2021). In addition, the Barcode of Life Data System (BOLD) database (http://www.boldsystems.org/, accessed date 24 May 2021) is used. GenBank and BOLD, containing the genetic data of millions of specimens, are in fact the reference public databases for the analysis of all organisms [32]. However, pitfalls in both these online depositories are reported, mainly related to the degree of taxon coverage they offer, and the reliability of the sequence-associated identification [33,34,35,36,37]. The need for a well-validated dataset containing accurate sequences has been strongly remarked upon for ensuring the proper identification of fungi [2,22,28,38], since the success of the analysis greatly depends on the technical quality and the correct taxonomic identification of the deposited sequences [28]. Therefore, a preliminary analysis (filtering) aimed at assessing the reliability of the official DNA databases and the creation of curated sequence databases is fundamental to achieve the model of fungi sequence-based identification [2,22]. In particular, the development of an ITS dataset targeting specific groups of mushrooms represents a widely used approach for a rapid identification via barcoding [25].

This study represents the first attempt at building and validating an ITS gene dataset for supporting the National Health Service in mushroom species identification, especially focusing on those responsible for poisoning cases.

## 2. Materials and Methods

### 2.1. Target Species: Selection Criteria

The following criteria were used for selecting the target species to be included in the ITS gene dataset: (1) Mushroom species involved in poisoning incidents in Tuscany during the ten-year period 2007 to 2017 and reported in a list provided by the Tuscany Poison Control Centre (Tus-PCC) (Table 1) (when only the genus was reported, all the species included in that genus and present in Tuscany were included). The correctness of the species nomenclature in the Tus-PCC list was verified by consulting the online database Index Fungorum (http://www.indexfungorum.org/, accessed date 24 May 2021), owned by the International Mycological Association and constantly updated with all the mycological nomenclatural novelties. Thus, for target species listed with obsolete or non-valid nomenclature, the valid name was used; (2) species morphologically similar to those reported in the Tus-PCC list and present in the Tuscany Region; and (3) edible mushroom species commonly collected/harvested and other mushroom species (edible or not) widely distributed in Tuscany. Points 2 and 3 were performed by an expert mycologist belonging to TMGA (https://www.agmtmicologia.org/, accessed date 24 May 2021) and mycological inspectorates, which also contributed to the categorization of the selected target species into edible (E), suspected not-edible (SNE), not-edible (NE), toxic (T), and mortal (M).

### 2.2. ITS Sequences Retrieval from Public Genetic Databases

For each target species, all the available ITS sequences were searched and retrieved from GenBank (https://www.ncbi.nlm.nih.gov/genbank/, accessed date 24 May 2021). Additional ITS sequences were retrieved from BOLD (http://www.boldsystems.org/, accessed date 24 May 2021), only if not reporting the info “mined from GenBank”, to avoid retrieving duplicates.

### 2.3. Systematic Double-Step ITS Sequences Filtering and Intra-Species Divergence Estimation

All sequences retrieved from the databases were systematically filtered. In detail, only sequences presenting pre-determined inclusion criteria (reported in the following sections) were maintained in the dataset, while the others were discarded.

#### 2.3.1. First Step: Sequence Quality and Technical Check

Sequences were discarded if: (1) declared as “unverified”; (2) reporting biological naming conventions between the genus name and the species name such as “aff.” (abbreviation for *affinis*) or “cf.” (abbreviation for *confer*), which are qualifiers used in taxonomy to indicate different degrees of uncertainty of identification [39]; (3) not including both the ITS-1 and ITS-2 regions; (4) shorter than 550 bp in length; since the ITS region length varies from ~600–800 bp in fungi [30], 550 bp was considered as the limit length under which sequences might result in being not informative enough; and (5) presenting IUPAC degenerate base symbols (over 4 N, B, D, H, and V and over 10 R, Y, S, W, K, and M) or homopolymers runs errors, that are associated with a scarce sequence quality. In this phase, sequences submitted with obsolete names were renamed with valid ones. Sequences belonging to species variants were attributed to the species level.

#### 2.3.2. Second Step: Phylogenetic Analysis

All the sequences retained after the first filtering step were aligned with Bioedit 7.0 software [40]. A phylogenetic analysis was conducted in MEGA7 [41] by using the Neighbor Joining (NJ) method based on the Kimura 2 parameter model [42] with 1000 bootstrap replicates. The NJ method was selected as used in GenBank and BOLD for producing distance trees of results. Clusters with bootstrap values ≥ 70% were considered as strongly supported [43]. Sequences clustering with species or genus different from the respective cluster (with bootstrap values ≥ 70%) were queried against the Basic Local Analysis Search Tool (BLAST) on GenBank. They were discarded if they showed identity values ≥ 97–100% [25] with species different to the one declared.

#### 2.3.3. Intra-Species Divergence Estimation

For each target species (for which more than one ITS sequence was maintained after the double-filtering process), the intra-species mean divergence was estimated using MEGA7 software [41].

### 2.4. Samples Collection, Morphological Identification, and Production of ITS Internal Reference Sequences

#### 2.4.1. Mushroom Samples Collection and Morphological Identification

The collection of fresh mushroom samples was performed during 2019 and 2020 by the TMGA in Tuscan territory. Only specimens from target species of the ITS gene dataset were collected. The collection was mainly focused on species for which no sequences or only one sequence was available after the ITS sequences’ systematic filtering (Section 2.3), for increasing the ITS gene dataset taxonomic coverage, and on species included in the Tus-PCC list, more commonly involved in poisoning cases. All the samples were morphologically identified by the TMGA. The specimens were photographed in situ. Info on the specimens’ collection sites and growth substrate were also annotated. Collected specimens were registered with an internal code and stored at −20 °C until further analysis.

#### 2.4.2. Total DNA Extraction and Evaluation

Total DNA extraction was performed starting from ~20 mg of tissue following the protocol described by Armani et al. [44]. The quality and quantity of the DNA extracted from each sample were determined with a NanoDrop ND-1000 spectrophotometer (NanoDrop Technologies, Wilmington, DE, US). Each DNA sample was stored at −20 °C until further analysis.

#### 2.4.3. ITS Region Amplification, Sequencing, and Sequence Editing

The ITS region was amplified with the primers ITS1 (5′-TCCGTAGGTGAACCTGCG-3′) and ITS4-B (5′-CAGGAGACTTGTACACGGTCCAG-3′) [30,45] using the following PCR protocol: 20 μL reaction volume containing 2 μL of a 10× buffer (BiotechRabbit GmbH, Berlin, Germany), 100 mM of each dNTP (Euroclone Spa, Milano, Italy), 200 nM of forward primer, 200 nM of reverse primer, 1.0 U PerfectTaq DNA Polymerase (BiotechRabbit GmbH, Berlin, Germany), 100 ng of DNA, and DNase free water (Euroclone Spa, Milano, Italy). The following cycling program was applied: denaturation at 95 °C for 3 min; 35 cycles at 95 °C for 30 s, 54 °C for 30 s, and 72 °C for 30 s; and final extension at 72 °C for 7 min. Five microliters of each PCR product was checked by gel electrophoresis on a 2% agarose gel. The amplification of fragments of the expected length was assessed by making a comparison with the standard marker SharpMass™ 50-DNA ladder (Euroclone Spa, Milano, Italy), and the concentration of PCR products by making a comparison with the intensity of the bands of the DNA ladder. Positive reactions were sequenced. Chromatograms were checked, searching for putative reading errors, and these were corrected. The ITS reference sequences were deposited in GenBank (http://www.ncbi.nlm.nih.gov/genbank/, accessed date 24 May 2021).

### 2.5. Building of the Final ITS Gene Dataset

#### 2.5.1. Taxonomic Coverage

The ITS reference sequences obtained in this study from specimens belonging to species with no sequences available after the double-step filtering process were included in the ITS gene dataset constructed in Section 2.3 and the overall taxonomic coverage (number of species finally present in the database/number of target species) was calculated. The final ITS gene dataset was uploaded and used in Geneious Prime version 2020.0.4 [46].

#### 2.5.2. Evaluation of ITS Region Efficiency in Species Identification

The ability of the ITS region to discriminate among target species included in the ITS gene dataset—and especially those causing poisonings at the national level (Tus-PCC list)—was evaluated according to Badotti et al. [28], who proposed an extensive comparative analysis based on the probability of correct identification of all Basidiomycota sequences deposited in GenBank using a selectively filtered dataset. Accordingly, they classified the ITS performance in species identification for genera of Basidiomycota into the following three distinct categories: good, intermediate, or poor [28].

### 2.6. ITS Gene Dataset Validation

For the validation process, all the reference ITS sequences obtained from this study (except those used to set up the ITS gene dataset in Section 2.5) were submitted to a BLAST analysis against the ITS gene dataset in Geneious Prime version 2020.0.4 [46]. An identity value ≥ 97–100% was selected as the threshold for species identification [25]. Results were discussed according to a comparison with outcomes from Section 2.5.

## 3. Results and Discussion

### 3.1. Target Species Selection

Overall, 242 mushroom target species belonging to 59 genera, 23 families, 7 orders, 2 classes, and 2 phyla (Basidiomycota and Ascomycota) were selected for building the ITS gene dataset (Appendix A). Overall, 98 species (40.5%) were classified as E, 24 (9.9%) SNE, 32 (13.2%) NE, 74 (30.6%) T, and 14 (5.8%) M (Appendix A). Basidiomycota were more represented, including 96.3% of the species (n = 233). This phylum is the second largest in the fungi kingdom and comprises approximately 30% of all described fungal species [47].

The Tus-PCC list reported 448 poisoning cases (Table 1), involving 41 species (78.8%) and 11 genera (21.2%). These were related to 38 poisoning cases (8.5%), highlighting the limits of the phenotypic identification during human intoxication. Often, mycologists are called to analyze clinical samples or leftovers of meals, where the key features of the whole specimens may lack. Moreover, despite the data referring to a recent past, obsolete names were provided for 9 of the 41 mushrooms species (21.9%) (Table 1). This particularly involved the genus *Boletus*, with three species—*B. lupinus*, *B. luridus*, and *B. pulchrotinctus*—re-classified as *Rubroboletus lupinus*, *Suillellus luridus*, and *S. pulchrotinctus* (Table 1). In fact, recent phylogenetic studies led to major taxonomic rearrangements of the Boletaceae family, introducing several new genera and leaving in the genus *Boletus*, in its new restricted sense, only *B. edulis* and closely related species [48].

Among the 41 species responsible for poisoning, 26 (63.4%) are known as T, 4 SNE (9.7%), 3 M (7.3%), and 1 not-edible NE (2.4%). However, seven species (17%) are commonly consumed and considered E.

Overall, a total of 109 species (all Basidiomycota) were selected as targets for the ITS gene dataset from the Tus-PCC list: the 41 reported at the species level and 68 species (toxic or not, and present in the territory) included in the 11 genera. In detail, 21 species were included for *Russula* spp., 4 for *Inocybe* spp., 6 for *Lepiota* spp., 3 for *Clitocybe* spp., 2 for *Entoloma* spp., 4 for *Macrolepiota* spp., 1 for *Panaeolus* spp., 6 for *Ramaria* spp., 8 for *Agaricus* spp., 11 for *Amanita* spp., and 2 for *Rhodocollybia* spp. (Appendix A). The remaining 133 target species were selected by the criteria reported in points 2 and 3 in Section 2.1.

### 3.2. ITS Sequence Retrieval from Public Databases

Overall, 7270 sequences were found, of which 6715 (92.4%) were found in GenBank and 555 (7.6%) in BOLD systems (Figure 1; Appendix A). Many sequence records appear in both the databases, which are intended to be complementary; in fact, the sequences submitted independently to GenBank are transmitted to BOLD periodically, while records from BOLD are submitted to GenBank when they are published [37]. In this phase, a taxonomic coverage of 93.8% was observed, with ITS sequences available for 227 out of the 242 target species. GenBank showed a taxonomic coverage of 92.6%, with sequences available for 224 species. The taxonomic coverage of BOLD could not be calculated, as the GenBank duplicate sequences were not considered (Section 2.2). However, absence of ITS sequences in this database was observed for many species during the collection phase (data not shown). BOLD factually poorly contributed to the overall taxonomic coverage (only three additional species). This evidence confirms the fact that this genetic database, although containing more curated data, is still not adequately populated [28]. In this respect, the BOLD handbook reported that the number of fungal sequences is relatively limited compared to the number of animal sequences, and thus a successful species match can be improved only if new sequences are added to the database (https://v3.boldsystems.org/index.php/resources/handbook?chapter=2_databases.html, accessed date 24 May 2021).

Cases of sequences deposited with obsolete names were observed. As also mentioned in Section 3.1, this aspect especially involved the re-classification of some *Boletus* spp. in other genera (*Caloboletus* spp.; *Neoboletus* spp., *Rubroboletus* spp.; *Suillellus* spp., *Suillus* spp., etc.).

### 3.3. Systematic Double-Step ITS Sequences Filtering, Intra-Species Divergence Estimation, and Evaluation of ITS Region Efficiency

#### 3.3.1. First Step: Sequences Quality and Technical Check

This step allowed us to discard 1087 sequences (14.9%), reducing the number of sequences to 6183 (Figure 1; Appendix A). All the inclusion criteria of this step were also considered in the ITS data filtering process performed by Badotti et al. [28], together with the check for the ITS sequences from permanent collections whose taxonomic identifications were curated by specialists (voucher specimens). Although the need for sequences to be associated with accurate specimen data and current species names had already been suggested [22], we decided to also include sequences from non-voucher specimens for assessing the actual data reliability within both public databases and also with the aim to expand the species taxonomic coverage of the dataset. Indeed, several cases of poor-quality sequences belonging to voucher specimens were highlighted in both GenBank and BOLD during the technical check. A number of sequences deposited with obsolete, erroneous, or imprecise names, so-called “dark taxa” [49], were encountered in both databases. This was more understandable in GenBank, since it essentially relies on users to accurately name their sequences [22].

#### 3.3.2. Second Step: Phylogenetic Analysis

During the sequence alignment, one not-aligning sequence was found. The BLAST analysis proved that this sequence, deposited as *Ramaria botrytis* ITS (GenBank accession n.: KY626150), presented identity values > 99% with sequences of the *R. botrytis* large subunit RNA ribosomal region, proving it was wrongly deposited. The sequence was discarded, and the subsequent analysis was performed on the 6182 remaining sequences. In this step, 205 sequences (3.3%) clustered separately from the respective species/genus with bootstrap values ≥ 70% (some examples were given in Figure 2). Results from the BLAST analysis of these sequences show identity values ≥ 97–100%, mostly with other co-generic species (n = 168; 81.9%), e.g., the sequence AJ131127, deposited as *Agaricus xanthodermus*, showed 100% identity values with sequences of *A. arvensis* and ≤ 89.3% with all the *A. xanthodermus* sequences. Additionally, cases of matching with species from other genera were found (n = 37; 18%), e.g., the sequence MH855192, deposited as *Lacrymaria lacrymabunda*, showed > 99% identity values with *Psathyrella candolleana* and ≤ 84.1% with all the *L. lacrymabunda* sequences. As observed in Section 3.3.1, sequences from voucher specimens were also involved. Thus, these 205 sequences were discarded and 5977 sequences (Figure 1), belonging to 224 species from 58 genera, 23 families, 7 orders, 2 classes, and 2 phyla, were finally maintained in the ITS gene dataset. In this phase, the dataset showed an overall species coverage of 92.6% (224 out of the 242 target species) and included 12 (5.4%) mortal M, 72 (32.1%) T, 32 (14.3%) NE, 21 (9.4%) SNE, and 87 (38.8%) E species. The Basidiomycota order alone included 233 species belonging to 56 genera.

Overall, 1293 sequences (17.8%) were discarded at the end of the systematic filtering process (Figure 1). Both sequences from GenBank and BOLD were involved, despite the latter being commonly considered as more reliable, containing sequences that surely belong to vouchered specimens (sequences must in fact fulfil some requirements, such as voucher data, collection record, and trace files that are instead not mandatory in GenBank) [28]. Issues in public databases have been already reported: Nilsson et al. [50] highlighted a certain lack of reference libraries’ accuracy, referring to the fact that about 20% of the entries deposited in GenBank were incorrectly identified to the species level, and that the majority of entries lacked descriptive and up-to-date annotations, especially regarding the taxonomic names; Schoch et al. [29] estimated that only approximately 50% of the ITS sequences that are deposited in public databases are annotated at the species level. Despite these considerations dating back to a decade ago, many of the mentioned issues still affect public databases.

#### 3.3.3. Intra-Species Divergence Estimation

The intra-species divergence was calculated for 209 species (5962 ITS sequences) out of the 224, since species with only one available sequence (n = 15) were not considered. The average intra-specific variability was 3.46% (±6.31) (Appendix A). Values ranged from 1 to 3% for most of the species (n = 95; 45.5%), followed by 0% (n = 58; 27.7%), 4–9% (n = 37; 17.7%), and ≥10% (n = 19; 9%). Four out of the five species from *Inocybe* spp. showed ≥ 10% average intra-specific variability (Appendix A). In the past decades, the average intra-specific ITS variability in mushrooms was estimated around 3%, with a standard deviation of 5.62% [51,52]. The average weighted intra-specific ITS variability was calculated as 3.33% (±5.62%) for Basidiomycota and 1.96% (±3.73%) for Ascomycota [53]; however, it was underlined that these values require further evaluation and constant updating [52].

Considering that in our study most of the target species belonged to Basidiomycota, the intra-species divergence value appeared in line with the values observed for this order [53].

### 3.4. Samples Collection, Molecular Analysis, and Production of ITS Internal Reference Sequences

#### 3.4.1. Mushroom Sample Collection

Overall, 97 specimens from 76 target species (all Basidiomycota) were collected by the TMGA (Table 2), of which 31 (40.8%) were T, 25 (32.9%) E, 11 (14.5%) NE, 5 (6.6%) SNE, and 4 (5.3%) M. Sampling from this study confirmed that misidentification may occur when toxic mushrooms appear like edible ones [16]. *Tricholoma sejunctum* (T), for example, can look identical to *T. scalpturatum* (E)*,* in shape, form, and color (Figure 3). Fifteen specimens from 10 species (*Amanita caesaria, Cantherellus ferruginascens, Cortinarius semisanguifluus, Desarmillaria tabescens, Macrolepiota venenata, Russula vinosobrunnea, Tricholoma basirubens, T. bresadolanum, T. gausapatum,* and *T. quercicola*) out of the 18 for which no sequences were available after the systematic filtering process were collected. For the remaining eight species (Appendix A), the collection was hindered by time limits and by difficulties in the specimen retrieval due to the specific picking season. The dataset will be further implemented in the future (see Section 3.6). Additionally, 12 specimens from 6 species (*Amanita excelsa, Infundibulicybe mediterranea, Lepista glaucocana, Russula nobilis, R. torulosa,* and *Tricholoma fracticum*) were collected. Twenty-nine species out of the 76 collected (38.2%) were among those selected from the Tus-PCC list (Section 3.1). In particular, almost all the species involved in the highest number of poisonings (Table 1) were collected: *Entoloma sinuatum, Omphalotus olearius, Russula* spp. (we collected *R. caerulea* (SNE)*, R. heterophylla* (E)*, R. nobilis* (NE)*, R. persicina* (NE)*, R. queletii* (T)*, R. torulosa* (T)*, R. vinosobrunnea* (E)), *Rubroboletus satanas, Agaricus xanthodermus, Macrolepiota venenata, Amanita phalloides, A. muscaria, Inocybe* spp. (we collected *I. geophylla* (T)), and *Lepiota* spp. (we collected *L. brunneoincarnata* (M)*, L. clypeolaria* (T), and *L. ignivolvata* (T)).

#### 3.4.2. DNA Extraction, ITS Region Amplification, and ITS Reference Sequences Production

The extraction protocol proposed by Armani et al. [44], although developed on seafood, was proven effective on fungal matrices. A good DNA quality and concentration was observed in all the samples, as the spectrophotometric analysis confirmed medium-high yield and quality (A260/A280 and A260/A230 ratio > 2.0) for all the collected samples (data not shown). The target ITS region was successfully amplified from all the DNA samples. The primer pair ITS1 and ITS4-B was selected since the primer ITS4-B [30], when paired with the universal primer ITS1 [45], efficiently amplifies DNA from all Basidiomycetes and Ascomycetes [30]. All the ITS amplicons were successfully sequenced. Overall, 97 sequences belonging to 76 species were produced (Table 2). After the editing phase, the sequences’ length ranged from 591 to 772 bp, in line with the length range reported by Gardes and Bruns [30]. 

The production of ITS reference sequences of species for which no sequences or only one sequence was available in the databases after the systematic filtering process allow us to calculate the intra-species overall mean divergence of an additional 10 species. In particular, the average intra-specific variability was 0% for *Cantharellus ferruginascens*, *Cortinarius semisanguifluus*, *Infundibulicybe mediterranea*, *Macrolepiota venenata*, and *Tricholoma gausapatum*, 1% from *T. fracticum*, 2% for *Russula nobilis* and *R. torulosa*, 4% for *Lepista glaucocana*, and 27% for *Amanita excelsa*. This latter result looks weird, especially considering that the observed average intra-specific variability of *Amanita* spp. ranged from 0 to 7% (Appendix A). Given the low number of available ITS sequences for these additional 10 species (2–4 sequences each), these outcomes should be carefully considered, and the intra-species divergence should be further investigated.

### 3.5. Building of the Final ITS Gene Dataset

Of the 97 reference sequences, the 15 produced from the 10 species (Section 3.4.2) for which no sequences were available after the systematic filtering process were included in the ITS gene dataset for implementing its taxonomic coverage, while the other 82 (belonging to 66 species) were used for the validation phase (Figure 1; Table 2).

#### 3.5.1. Taxonomic Coverage Implementation of the ITS Gene Dataset

The ITS gene dataset expanded with 15 reference sequences from 10 species showed an overall taxonomic coverage of 96.7%, including 234 out of the 242 target species. It finally contained 13 (5.6%) M, 74 (31.6%) T, 32 (13.7%) NE, 22 (9.4%) SNE, and 93 (39.7%) E species.

#### 3.5.2. Evaluation of ITS Region Efficiency in Species Identification

Although the ITS region is not equally variable in all groups of fungi [28,52,54,55], most fungal species have been identified based on this region and, consequently, most available sequences belong to this commonly used marker [28,55]. This means that, regardless of several limitations, the ITS region will likely remain the main marker of choice for fungal identification in the immediate future [22]. For these reasons, it was selected as a marker in this study.

As a general rule, a species was considered successfully identified if the minimum inter-specific distance was larger than its maximum intra-specific distance [56]. Applying the categories suggested by Badotti et al. [28] to the target species of this study, the ITS region was considered as a good marker for 19 out of the 56 (32.7%) Basidiomycota target genera considered in this study (Appendix A). For the other 10 (17.8%) and 2 (3.6%) genera, the identification performance was considered as intermediate and poor, respectively. The ITS species identification performance was instead not evaluated by Badotti et al. [28] for 23 genera (41.1%) herein included (Appendix A). This is probably due to the continuously evolving re-classification and allocation of mushroom genera.

Despite the ITS region being proven as a good marker only for a moderate percentage of the target genera included in this study, it should be noted that these included a plurality of the target species (n = 110; 47.2%); contrariwise, a lower number of species belonged to the genera included in the intermediate (n = 53; 22.7%) and poor (n = 18; 7.7%) categories (Appendix A). Additionally, for most of the 109 species selected from the Tus-PCC list (Section 3.1) (n = 75; 68.8%) the ITS region can be considered as a good discrimination marker. For the remaining species, ITS performance was classified as intermediate in 18 species (16.5%), poor in only 1 species (0.9%), and not evaluated in 15 species (13.8%). Thus, especially considering the use for which the ITS gene dataset is intended, namely a tool for supporting the species identification in poisoning cases, we considered these findings particularly promising. To confirm this, a validation process was performed, as reported in the following section.

### 3.6. Validation of the Final ITS Gene Dataset

The ITS dataset validation process was performed in Geneious Prime version 2020.0.4 [38] by BLAST analysis of 82 reference sequences from 66 species (19 E, 4 SNE, 11 NE, 29 T, and 3 M) (Table 2) which remained after removing the 15 sequences from 10 species used to improve the dataset taxonomic coverage (Section 3.5.1). Doing this, we also validate mushroom identifications by comparing molecular results with morphological identifications performed by an expert mycologist. In fact, only properly identified and labeled sequences should be used as references for accurate fungal identification [57].

In most cases (n = 71; 86.6%), the reference sequences properly matched with identity values ≥ 97–100% uniquely with the corresponding species. Five reference sequences (6.1%) matched uniquely with the corresponding species but with identity values < 97% (Table 2). This could not be explained by the observed average intra-species variability, ranging from 0 to 1% (Appendix A). Given the low number of sequences available for these species (Appendix A), the estimated intra-species mean divergence may not always be reliable, and more reference sequences should be produced. Additionally, four sequences (4.9%) simultaneously matched with the corresponding species and with others, with identity values ≥ 98%. The two sequences of *A. excelsa* showed higher identity values with *A. pantherina* than with the only available co-specific sequence. These two sequences were not deposited in GenBank. In fact, also considering the observed intra-species variability (Section 3.4.2), further investigation is needed. According to these outcomes, 55 out of the 66 species (83.3%) used to validate the ITS gene dataset were unequivocally allocated to a species. These included all the M and NE species and almost all the T (28 out of 29) and SNE (3 out of 4) species (Figure 4).

At the end of the validation, 95 sequences from 75 species were deposited in GenBank (GenBank accession numbers MZ005473-MZ005566).

Despite the validation process being successful for most of these species, it is appropriate to observe that the ITS gene dataset is not 100% effective in identifying mushroom species. As already discussed, this is attributable to some limits of the genetic marker itself. Thus, the use of a multigene analysis relying on comparison of at least three loci had already been proposed for fungi identification [58].

However, by putting together the outcomes from the validation process and those from the evaluation of ITS region efficiency (Section 3.5.2), the ITS gene dataset was proved as particularly efficient in identifying the species mostly involved in poisoning cases. Overall, 373 out of the 448 poisoning cases (83.2%) involved T species; most of them (n = 310; 83.1%) referred to *Entoloma sinuatum*, *Omphalotus olearius*, and *Clitocybe nebularis* (Table 1). Eleven poisoning cases (2.9%) were related to M species, mostly (n = 7; 63.3%) to the well-known *Amanita phalloides*, which is one of the most common poisonous mushrooms, responsible for 90% of human fatal cases globally [59]. However, 15 poisoning cases (4%) were even related to E species, especially *Armillaria mellea* (n = 6; 40%), confirming the potential toxicity of any mushroom in certain circumstances [12,14]. This aspect highlights that the control system cannot only rely on the knowledge of a restricted group of “popular” toxic species, but it must necessarily take into consideration any possible mushroom species present in the territory for guaranteeing its effectiveness. Interestingly, many analogies were observed by comparing the Tus-PCC list with the national poisoning cases, as the involved species generally overlap, and the major number of poisonings are caused by *E. sinuatum*, *O. olearius*, and *C. nebularis* also at the national level [17,18]. Likewise, for the most part, the edible species that were proved able to cause poisonings with gastrointestinal syndrome at the national level are the same as of the Tus-PCC list, with *A. mellea* and allied species ranking first [17]. Many of the mentioned species were also frequently involved in poisoning episodes in other EU countries, together with other strictly local species [11,15,60,61,62].

Therefore, the ITS gene dataset, opportunely integrated, may also be used by official laboratories at both the national and European level, which only have to include sequences from reference local species for guaranteeing the territorial coverage.

## 4. Conclusions

The ITS gene dataset built in this study represents the first attempt at collecting reliable data that could support mushroom species identification, especially for species responsible for poisoning. Its versatility, related to the possibility to constantly update the data, and consequently increase the taxonomic coverage and expand the application field from time to time, undoubtedly represent a strength. In addition, the availability of a dataset “depurated” from erroneous sequences would accelerate the interpretation of the results obtained after the query of official samples. In this light, the ITS gene dataset can represent a valid tool for the National Health Service to quickly react to poisoning incidents by the analysis of clinical samples. This is crucial, as a proper detection of the species involved in the poisoning can guide the most appropriate medical treatment. Moreover, the database can support the official control activities aimed at verifying the actual market condition, detecting fraud incidents, and safeguarding consumers. While many studies aimed at food authentication, in particular seafood, are available [63], few studies applying DNA barcoding for authentication of commercial mushroom products sold on the market have been conducted so far. Raja et al. [25] analyzed mushrooms used as food and/or dietary supplement, while Jensen-Vargas et al. [64] analyzed dried and fresh fungi that were sold in New York City supermarkets. Another recent survey analyzed more than 3500 mushroom samples collected in 35 countries across Yunnan Province [65]. This scarcity of studies could be due to the well-known difficulties in identifying mushrooms species such as, among others, the fact that the correct identification is entirely dependent on the availability of reliable sequences in public databases, the capacity of correct interpretation of BLAST results, and the resolution of the ITS marker alone for a precise identification [35]. It should be noted that, besides its use in standard sequencing methods (DNA barcoding), the ITS gene dataset can also be used in metabarcoding approaches related to high-throughput sequencing technologies, which are especially useful for food authentication from complex food matrices. The ITS gene dataset development especially fits the requirement of Article 98 of Regulation (EU) No. 2017/625 [66] addressing the responsibilities and tasks of European Union reference centers for the authenticity and integrity of the agri-food chain, which shall be responsible for different tasks, among which, where necessary, is establishing and maintaining collections or databases of authenticated reference materials.

## Figures and Tables

**Figure 1 foods-10-01193-f001:**
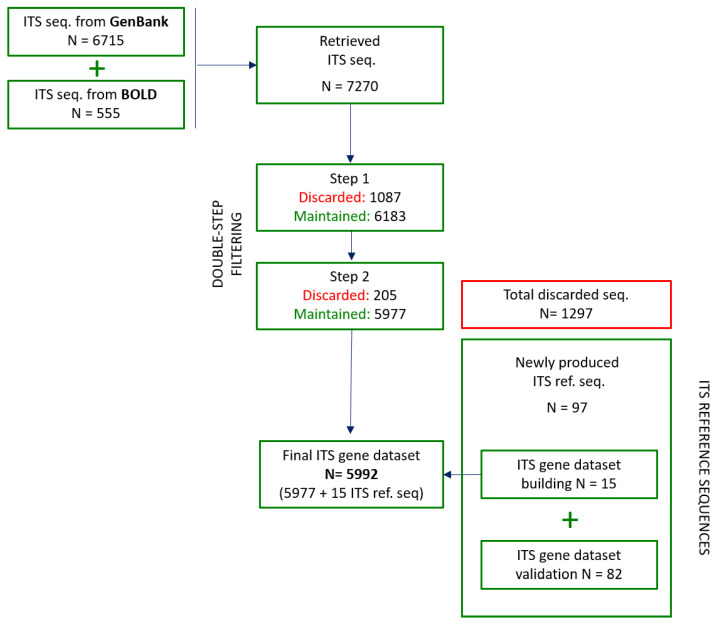
Diagram of the ITS gene dataset building process. The ITS sequences retrieved from public databases and their systematic double-step filtering process, as well as the ITS reference sequence production, are reported.

**Figure 2 foods-10-01193-f002:**
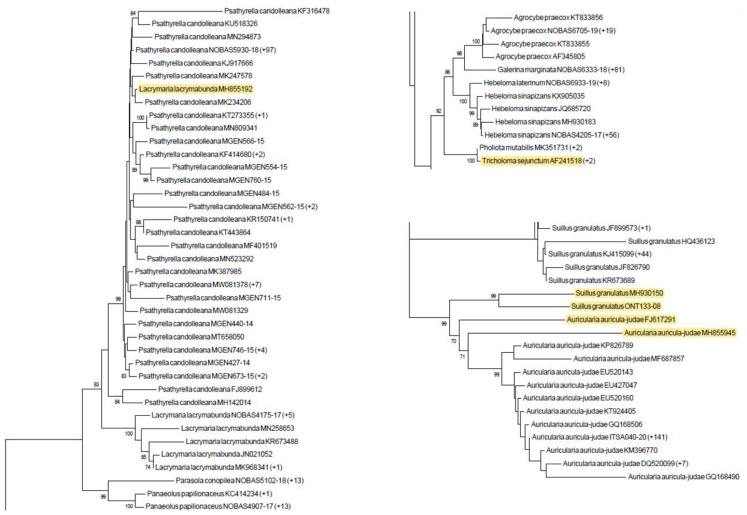
Three sections of the Neighbor Joining (NJ) phylogram constructed using the Kimura 2 parameter model [42] with 1000 bootstrap replicates on ITS sequences maintained after the first filtering step. Examples of sequences which clustered separately from the respective species/genus with bootstrap values ≥ 70% (highlighted in yellow) were therefore removed from the dataset. In brackets, the number of ITS sequences presenting an intra-species variability < 0.01 with respect to the indicated sequence is reported.

**Figure 3 foods-10-01193-f003:**
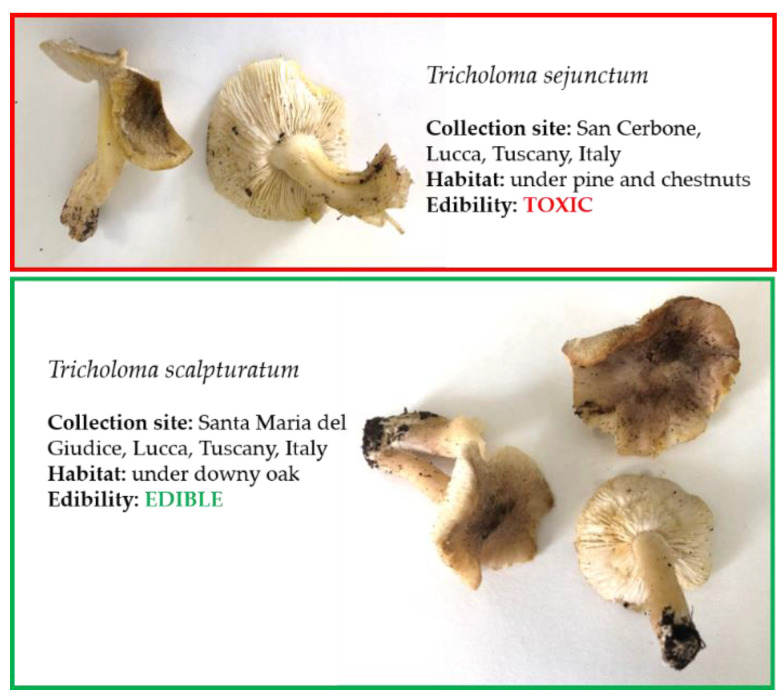
Pictures of collected specimens from *Tricholoma* spp. with relative collection site and habitat. Morphologic similarity among toxic (*T. sejunctum*) and edible (*T. scalpturatum*) species can be observed.

**Figure 4 foods-10-01193-f004:**
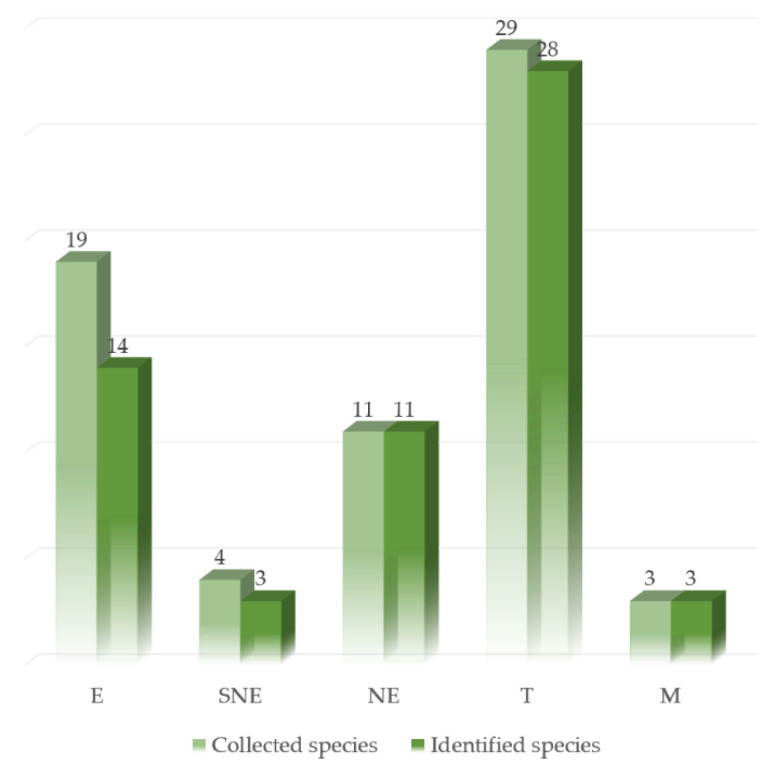
Comparison of the number of collected species used to validate the ITS gene dataset and the number of species unequivocally identified, divided into edibility categories. E: edible; SNE: suspected not edible; NE: not edible; T: toxic; M: mortal.

**Table 1 foods-10-01193-t001:** List of mushroom poisoning cases that occurred in Tuscany during the ten-year period 2007 to 2017. The number of cases and the involved species/genus are reported. Obsolete nomenclatures are integrated with currently valid scientific names (in brackets) according to Index Fungorum (http://www.indexfungorum.org/, accessed date 24 May 2021). The species edibility is also indicated. E: edible; SNE: suspected not edible; NE: not edible; T: toxic; M: mortal; * more than one species included in different edibility categories.

Cases (n)	Species/Genus	Edibility
153	*Entoloma sinuatum*	T
135	*Omphalotus olearius*	T
22	*Clitocybe nebularis*	T
13	*Russula* spp.	*
10	*Boletus satanas (Rubroboletus satanas)*	T
9	*Agaricus xanthodermus*	T
9	*Macrolepiota venenata*	T
7	*Amanita phalloides*	M
7	*Macrolepiota rachodes (Chlorophyllum rhacodes)*	SNE
7	*Amanita muscaria/A. aureola (A. muscaria)*	T
6	*Armillara mellea*	E
5	*Inocybe* spp.	*
4	*Lepiota* spp.	*
3	*Amanita pantherina*	T
3	*Boletus luridus (Suillellus luridus)*	E
3	*Clitocybe* spp.	*
3	*Clitocybe rivulosa*	T
3	*Entoloma* spp.	*
3	*Lepiota josserandii/L. subincarnata (L. subincarnata)*	M
3	*Macrolepiota* spp.	*
3	*Psathyrella candolleana*	T
2	*Amanita ovoidea*	T
2	*Armillaria tabescens (Desarmillaria tabescens)*	E
2	*Lepiota cristata*	T
2	*Panaeolus* spp.	*
2	*Ramaria* spp.	*
2	*Tricholoma saponaceum*	T
1	*Agaricus* spp.	*
1	*Agaricus preclaresquamosus (Agaricus moelleri)*	T
1	*Amanita* spp.	*
1	*Amanita proxima*	T
1	*Amanita verna*	M
1	*Boletus lupinus (Rubroboletus lupinus)*	T
1	*Boletus pulchrotinctus (Suillellus pulchrotinctus)*	T
1	*Cantharellus cornucopioides*	E
1	*Clitocybe dealbata*	T
1	*Entoloma lividoalbum*	SNE
1	*Entoloma rhodopolium*	SNE
1	*Hygrophoropsis aurantiaca*	SNE
1	*Hypholoma fasciculare*	T
1	*Inocybe rimosa*	T
1	*Lactarius zonarius*	T
1	*Lepista nuda*	E
1	*Ramaria flavescens*	E
1	*Ramaria formosa*	T
1	*Ramaria pallida*	T
1	*Rhodocollybia* spp.	*
1	*Russula foetens*	T
1	*Russula persicina*	NE
1	*Russula torulosa*	T
1	*Suillus collinitus*	E
1	*Tricholoma equestre*	T
Total: 448		

**Table 2 foods-10-01193-t002:** Samples collected in this study (species and specimens number) and obtained ITS reference sequences used to set up and validate the ITS gene dataset. * sequences matching uniquely with the corresponding species but with identity values < 97%; ** sequences simultaneously matching with the corresponding species and one other of the same edibility, with identity values ≥ 98%.

	Species	Edibility	Specimens (n)	ITS Ref. Seq. (n)
building	*Amanita caesarea*	E	1	1
	*Cantharellus ferruginascens*	E	2	2
	*Cortinarius semisanguifluus*	M	3	3
	*Desarmillaria tabescens*	E	1	1
	*Macrolepiota venenata*	T	2	2
	*Russula vinosobrunnea*	E	1	1
	*Tricholoma basirubens*	E	1	1
	*Tricholoma bresadolanum*	T	1	1
	*Tricholoma gausapatum*	E	2	2
	*Tricholoma quercicola*	SNE	1	1
TOT.	10		15	15
validation	*Agaricus bresadolanus*	T	2	2
	*Agaricus menieri*	T	1	1
	*Amanita citrina*	T	1	1
	*Amanita excelsa*	SNE	2	2
	*Amanita muscaria*	T	1	1
	*Amanita ovoidea*	T	1	1
	*Amanita pantherina*	T	1	1
	*Amanita phalloides*	M	1	1
	*Amanita strobiliformis*	E	1	1 *
	*Boletus edulis*	E	2	2
	*Butyriboletus pseudoregius*	E	1	1
	*Clitocybe phaeophtalma*	T	1	1
	*Clitopaxillus alexandri*	SNE	2	2 *
	*Clitopilus prunulus*	E	1	1
	*Cortinarius cedretorum*	NE	1	1
	*Cortinarius orellanus*	M	1	1
	*Cyclocybe cylindracea*	E	1	1
	*Entoloma lividoalbum*	SNE	2	2
	*Entoloma sinuatum*	T	1	1
	*Gymnopus erythropus*	NE	1	1
	*Gyroporus castaneus*	T	1	1
	*Infundibulicybe geotropa*	E	1	1
	*Infundibulicybe mediterranea*	E	3	3
	*Inocybe geophylla*	T	1	1
	*Lactarius blennius*	T	1	1
	*Lactarius chrysorreus*	NE	1	1
	*Lactarius decipiens*	NE	1	1
	*Lactarius pyrogalus*	NE	1	1
	*Lactarius quietus*	NE	1	1
	*Lactarius torminosus*	T	1	1
	*Lactarius zonarius*	T	1	1
	*Lepiota brunneoincarnata*	M	1	1
	*Lepiota cristata*	T	1	1
	*Lepiota clypeolaria*	T	2	2
	*Lepiota ignivolvata*	T	1	1
	*Lepista glaucocana*	E	1	1 **
	*Lepista nuda*	E	1	1
	*Lepista sordida*	NE	1	1
	*Leucoagaricus leucothites*	T	1	1
	*Macrolepiota permixta*	E	1	1 **
	*Macrolepiota procera*	E	1	1
	*Neoboletus erythropus*	E	1	1 **
	*Omphalotus olearius*	T	1	1
	*Parasola conopilea*	T	2	2
	*Pholiota mutabilis*	E	1	1 *
	*Ramaria flavescens*	E	2	2
	*Ramaria formosa*	T	1	1
	*Ramaria pallida*	T	1	1 *
	*Rubroboletus satanas*	T	1	1 **
	*Russula caerulea*	SNE	2	2
	*Russula chloroides*	NE	1	1
	*Russula heterophylla*	E	1	1
	*Russula nobilis*	NE	2	2
	*Russula persicina*	NE	2	2
	*Russula queletii*	T	1	1
	*Russula sanguinea*	NE	1	1
	*Russula torulosa*	T	2	2
	*Russula vesca*	E	1	1
	*Scleroderma polyrhizum*	T	1	1
	*Suillus grevillei*	E	1	1
	*Suillus viscidus*	E	1	1
	*Tricholoma fracticum*	T	2	2
	*Tricholoma scalpturatum*	E	1	1
	*Tricholoma sciodes*	T	2	2
	*Tricholoma sejunctum*	T	1	1
	*Tricholoma sulphureum*	T	1	1
TOTAL	66		82	82

## Data Availability

Data sharing not applicable.

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
