# Peer review of "Building of an Internal Transcribed Spacer (ITS) Gene Dataset to Support the Italian Health Service in Mushroom Identification"

_foods, 2021, doi:10.3390/foods10061193_

Round 1
Reviewer 1 Report
Dear editor,
Thank you for inviting me to review the manuscript titled “Building of an Internal Transcribed Spacer (ITS) gene dataset to support the Italian Health Service in mushrooms identification” to build an ITS gene dataset able to identify species involved in regional poisoning incidents that can be used for supporting the Health Service in the Italian territory. Moreover, it can support the official control activities aimed at detecting frauds and safeguarding consumers.
The manuscript structure and ideas were good and I believe it could provide important information on mushrooms authenticity and food chain safety. I definitely agree that this manuscript should continue to peer-review. I have in PDF minor doubts (highlighted) that I think the authors might wish to confirm.
Strengths of this study:
- The materials and methods are sound.
- The conclusion is made adequately according to their major findings.
Limitations of this study:
- Authors should mention, in the introduction section, other analytical methods.
- References: authors should look for new references on mushroom identification and authenticity of foods based on analytical parameters to justify and discuss their findings.

Author Response
Thank you for inviting me to review the manuscript titled “Building of an Internal Transcribed Spacer (ITS) gene dataset to support the Italian Health Service in mushrooms identification” to build an ITS gene dataset able to identify species involved in regional poisoning incidents that can be used for supporting the Health Service in the Italian territory. Moreover, it can support the official control activities aimed at detecting frauds and safeguarding consumers.
The manuscript structure and ideas were good and I believe it could provide important information on mushrooms authenticity and food chain safety. I definitely agree that this manuscript should continue to peer-review. I have in PDF minor doubts (highlighted) that I think the authors might wish to confirm.
Strengths of this study:
- The materials and methods are sound.
- The conclusion is made adequately according to their major findings.
Limitations of this study:
- Authors should mention, in the introduction section, other analytical methods.
- References: authors should look for new references on mushroom identification and authenticity of foods based on analytical parameters to justify and discuss their findings.
We sincerely thank the reviewer for appreciating our work. We also think that the availability of an Internal Transcribed Spacer (ITS) gene dataset could support the official control activities aimed at detecting frauds and safeguarding consumers. As regard the suggestions provided:
In the manuscript we voluntarily focused the attention on studies that used the same approach we used (DNA barcoding of ITS region) for mushroom identification. However, a specific section, regarding other analytical methods used for mushroom identification, has been added in the introduction. Since our study was not specifically addressed to commercial mushroom products, similar studies have been cited only in the conclusion section and beside to the ones already mentioned, the study of Zhang et al., 2021 was also mentioned. We did not dedicate a specific section to the discussion of this topic considering we are carrying on a study focused on species authentication in commercial mushroom products sold on the Italian market.
Reviewer 2 Report
Review of the manuscript “Building of an Internal Transcribed Spacer (ITS) gene dataset 2 to support the Italian Health Service in mushrooms identification” submitted to the journal of Foods. In this manuscript, the authors aimed to generate an ITS gene dataset to support the Italian Health Services in mushroom identification with a focus on those involved in regional poisoning. Authors carried out several steps to quality check and validate the genes retrieved from some reference public databases, and conducted some post-processing steps such as phylogenetic and molecular analysis. Although the topic looks interesting and can be used as a starting point for such a purpose especially for Italy, I believe the manuscript is quite partial and the work still requires some adjustment and effort. Overall, the English writing of the manuscript is pretty poor and it does not read well. The organization of the manuscript requires some revision. In addition, the filtering, quality check, and analysis should be smoothly presented and clearly discussed. I found multiple redundant texts in different sections, long sentences, and lots of inconsistencies in the presentation of the work. Here I point out a few of the cases among many:
- Authors talk about filtering almost everywhere in the abstract but it is not clear what filtering. Obviously it is important to mention in the abstract that the data is filtered and high quality but the way it is presented is very ambiguous.
- Do not start sentences with numbers
- L45: the global market was valued…
- L99-100: wrong sentencing, please revise
- Section 2.3.2: please provide the phylogenetic tree in supplementary figures
- Repeating the steps and a lot of redundant info in the result section that waere presented in the M&M before
- I am not sure if the sample collection should be presented in the results and discussion section
- I am not sure if this is related to the submission platform or authors, but I was not able to get any of the supplementary files mentioned at lines 541 and 542. All the results presented in the manuscript should be provided with the manuscript and the final gene set with its description, etc. should be supplemented with the manuscript. I was not able to access any of those, hence did many not evaluate them
- Probably authors copied and pasted the references from Google Scholar or some incomplete sources, please check the references and make sure they are complete. Lots of your references have incomplete and missing info. Please look for the ? and … marks
Author Response
Review of the manuscript “Building of an Internal Transcribed Spacer (ITS) gene dataset 2 to support the Italian Health Service in mushrooms identification” submitted to the journal of Foods. In this manuscript, the authors aimed to generate an ITS gene dataset to support the Italian Health Services in mushroom identification with a focus on those involved in regional poisoning. Authors carried out several steps to quality check and validate the genes retrieved from some reference public databases, and conducted some post-processing steps such as phylogenetic and molecular analysis. Although the topic looks interesting and can be used as a starting point for such a purpose especially for Italy, I believe the manuscript is quite partial and the work still requires some adjustment and effort. Overall, the English writing of the manuscript is pretty poor and it does not read well. The organization of the manuscript requires some revision. In addition, the filtering, quality check, and analysis should be smoothly presented and clearly discussed. I found multiple redundant texts in different sections, long sentences, and lots of inconsistencies in the presentation of the work.
Dear reviewer, thank you for your suggestions. The manuscript has been revised also trying to reduce redundancy and long sentences. In particular, some sentences previous reported in R&D section were anticipated in M&M to avoid repetitions. English style has been revised by a native speaker throughout the entire manuscript and the correction are tracked.
Here I point out a few of the cases among many:
Authors talk about filtering almost everywhere in the abstract but it is not clear what filtering. Obviously it is important to mention in the abstract that the data is filtered and high quality but the way it is presented is very ambiguous.
The filtering process is now better explained in the Introduction and M&M sections.
Do not start sentences with numbers
Done
L45: the global market was valued…
Done
L99-100: wrong sentencing, please revise.
The sentence was modified.
Section 2.3.2: please provide the phylogenetic tree in supplementary figures
Since the NJ phylogram appeared too complex as containing thousands of sequences, sections of the NJ phylogram were provided a Figure in the text (Figure 2) as suggested by R4.
Repeating the steps and a lot of redundant info in the result section that were presented in the M&M before
Done: some sentences previous reported in R&D section were anticipated in M&M to avoid repetitions.
I am not sure if the sample collection should be presented in the results and discussion section.
Since the specimens’ collection strategy was set up starting from the outcomes from the sequence filtering phase, we think that presenting sample collection in results and discussion section is more appropriate.
I am not sure if this is related to the submission platform or authors, but I was not able to get any of the supplementary files mentioned at lines 541 and 542. All the results presented in the manuscript should be provided with the manuscript and the final gene set with its description, etc. should be supplemented with the manuscript. I was not able to access any of those, hence did many not evaluate them.
Dear reviewer, all the files related to the manuscript, included the supplementary files, were submitted to the journal. However, we encountered the same problem when we tried to visualize the Supplementary Material by the link available on the pdf. Therefore, the issue is related to the submission platform and we will ask to the journal to solve it.
Probably authors copied and pasted the references from Google Scholar or some incomplete sources, please check the references and make sure they are complete. Lots of your references have incomplete and missing info. Please look for the ? and … marks
References have been checked and revised according to the Journal guidelines except for Schoch et al., 2014 and Nilsson et al., 2014 for which more than 50 authors contributed to the article.
Reviewer 3 Report
Authors describe a ITS dataset to identify fungi from Tuscany. This dataset will support the National Health Service. It is a very descriptive work, especially the section of results, which is sometimes looks like materials and methods. Therefore, only very specific readers will be interested in this work. However, the article is notoriously well written.
minor concerns:
line 247: add a full stop after (E) and delete the rest of the sentence as it is repeated in the text.
Line 267: taxonomic coverage of 93.8 %, 92.6% from Genbank, and where does the remaining 1.2% come from if the taxonomic coverage of BOLD syste, could not be calcualted?
Line 502: Amanita phalloides in italics.
Author Response
Authors describe a ITS dataset to identify fungi from Tuscany. This dataset will support the National Health Service. It is a very descriptive work, especially the section of results, which is sometimes looks like materials and methods. Therefore, only very specific readers will be interested in this work. However, the article is notoriously well written.
minor concerns:
line 247: add a full stop after (E) and delete the rest of the sentence as it is repeated in the text.
Done
Line 267: taxonomic coverage of 93.8 %, 92.6% from Genbank, and where does the remaining 1.2% come from if the taxonomic coverage of BOLD system, could not be calculated?
We meant that, since deduplicated sequences present in BOLD were not considered and only additional sequences were retrieved, the overall taxonomic coverage of this database was not evaluated.
Line 502: Amanita phalloides in italics.
Done
Reviewer 4 Report
In this work, the authors describe creating a database of ITS sequences for hundreds of species of mushroom-forming fungi to act as a reference for identifying poisonous species. I think that this endeavor is both interesting and extremely useful to its target audience, and that the authors have been rigorous and thorough in their pursuit of it. However, I believe some improvements could be made that will enhance the impact of the work and make it more interesting for the reader.
Major Points:
- I cannot recall reading a paper with essentially no figures. Adding some figures will help break up the text and help to draw the interest of prospective readers. Three such figures occurred to me.
- Photos! How about some pictures of mushrooms? The methods section (lines 180-181) says that the specimens were photographed in situ at the time of collection. Why not add some of these? I think it would be really interesting to show side-by-side photos of a poisonous and a non-poisonous species that appear morphologically very similar but can be distinguished by their ITS sequences.
- Phylogenetic Tree. This may get too big to include all of the species, so perhaps there is a subset to could be presented. It would be nice to see the distribution of the species, especially if you add symbols to indicate which species are poisonous.
- A pie chart. One of the key findings of the paper is that not all species can be reliably identified by their ITS sequence (lines 477 – 488). In addition, not all species are poisonous. I think a pie chart could be a nice way to visualize the overlap between the poisonous vs non-poisonous compared with the numbers of species that can be uniquely identified by their ITS sequence versus not.
- To make room for these figures, the text could be edited down quite a bit. There is much too much repetition between the Methods section and the Results section.
- You should expand the Introduction a little. There were two topics that I was interested in and thought could help make the paper more interesting and impactful.
- How much does knowing exactly which poisonous species that someone ate affect treatment options in the clinic? Are there examples where the treatment for mushroom poisoning could be counter-indicated depending on if they had ingested species A or species B?
- The work is very narrowly focused on mushrooms in Italy. I don’t know anything about the distribution of mushroom species. Would this database be useful to someone in Spain or Germany? How about in China or Australia?
- I am extremely sympathetic to the issue that scientists whose first language is not English have to write their papers in English. I hate to say this, but the writing could really benefit from an edit by a skilled, native English speaker.
Minor Points:
- Please define edible, suspected not-edible, not-edible, toxic, and mortal. One can assume that you should not eat a mushroom called toxic, but what criteria distinguish not-edible versus toxic? Or toxic versus mortal?
- One a related note, several of the species in Table 1 are listed as edible or suspected not-edible. Shouldn’t inclusion in that list automatically warrant upgrading those species to not-edible or worse?
- Include the sequences or primers ITS1 and ITS4-B (line 191). There is no sense in making others have to hunt for this information.
- In Table 2 in the PDF that I received, the accession numbers are all missing. Please make sure that the table is properly formatted.
- Line 456-457, the group that includes 47.2% can be called “a plurality”, but not “most”.
Author Response
In this work, the authors describe creating a database of ITS sequences for hundreds of species of mushroom-forming fungi to act as a reference for identifying poisonous species. I think that this endeavor is both interesting and extremely useful to its target audience, and that the authors have been rigorous and thorough in their pursuit of it. However, I believe some improvements could be made that will enhance the impact of the work and make it more interesting for the reader.
Dear reviewer, thank you for the positive feed backs on our manuscript. We have revised the manuscript according to your suggestions.
Major Points:
I cannot recall reading a paper with essentially no figures. Adding some figures will help break up the text and help to draw the interest of prospective readers. Three such figures occurred to me.
Photos! How about some pictures of mushrooms? The methods section (lines 180-181) says that the specimens were photographed in situ at the time of collection. Why not add some of these? I think it would be really interesting to show side-by-side photos of a poisonous and a non-poisonous species that appear morphologically very similar but can be distinguished by their ITS sequences.
As properly suggested by the reviewer, an explicative figure was added in the manuscript (Figure 3). Given the high number of collected species, only one evident case of morphological similarity between poisonous and non-poisonous species was reported.
Phylogenetic Tree. This may get too big to include all of the species, so perhaps there is a subset to could be presented. It would be nice to see the distribution of the species, especially if you add symbols to indicate which species are poisonous.
As suggested, sections of the NJ phylogram were provided a Figure in the text (Figure 2).
A pie chart. One of the key findings of the paper is that not all species can be reliably identified by their ITS sequence (lines 477 – 488). In addition, not all species are poisonous. I think a pie chart could be a nice way to visualize the overlap between the poisonous vs non-poisonous compared with the numbers of species that can be uniquely identified by their ITS sequence versus not.
A graphic was included in the manuscript as suggested by the reviewer (Figure 4).
To make room for these figures, the text could be edited down quite a bit. There is much too much repetition between the Methods section and the Results section.
To avoid repetition and reduce the text length some part of results and discussion have been moved to M&M section.
You should expand the Introduction a little. There were two topics that I was interested in and thought could help make the paper more interesting and impactful.
How much does knowing exactly which poisonous species that someone ate affect treatment options in the clinic? Are there examples where the treatment for mushroom poisoning could be counter-indicated depending on if they had ingested species A or species B?
We agree with the reviewer about the interest of this topic. However, it is outside of our expertise, and this is the reason why we did not face it in the manuscript. It is appropriate to highlight that As various toxins produce evidently severe symptoms and death, their timely and accurate identification is essential, critical, as well as valuable for subsequent clinical diagnosis and treatment and, in this respect, a reliable and rapid molecular analysis can be useful.
The work is very narrowly focused on mushrooms in Italy. I don’t know anything about the distribution of mushroom species. Would this database be useful to someone in Spain or Germany? How about in China or Australia?
Thank you for the suggestion. We added some references related to poisoning cases in other EU countries. As regards the extra-EU scenario, we are pursuing a project dealing with species identification of mushrooms imported from Third Countries. The info required by the reviewer will be included in that study in a more deepened way.
I am extremely sympathetic to the issue that scientists whose first language is not English have to write their papers in English. I hate to say this, but the writing could really benefit from an edit by a skilled, native English speaker.
English style has been revised by a native speaker throughout the entire manuscript and the correction are tracked.
Minor Points:
Please define edible, suspected not-edible, not-edible, toxic, and mortal. One can assume that you should not eat a mushroom called toxic, but what criteria distinguish not-edible versus toxic? Or toxic versus mortal?
The categorization in edible, suspected not-edible, not-edible, toxic, and mortal was performed by expert mycologists, some of them working at the mycological inspectorates belonging to the National Sanitary System, on the bases of their knowledge and skills, also taking into consideration the national and international poisoning cases and scientific literature. Here we report the references that we not included in the reference list:
Assisi F., Balestreri S., Galli R. (2008) - Funghi velenosi. Tossicologia, speciografia e prevenzione. Dalla Natura, Milano.
Assisi F., Della Puppa T., Moro P., Bonacina E: (2002) – Le Amanite e le loro tossine. Atti del 2° convegno Internazionale di Micotossicologia.Viterbo, 6-7 dicembre 2001. Pagine di Micologia, 17: 7-19.
Benjamin D. (2020) – La commestibilità dei funghi: miti e malintesi. Mushroom Edibility: Myths
and misunderstandings . 6° convegno Internazionale di Micotossicologia. Perugia 23-24 novembre 2018. Pagine di Micologia 41: 13-21.
Berndt S. (2020) – La sindrome "neurologica" da morchelle. Atti del 6° convegno Internazionale di Micotossicologia. Perugia 23-24 novembre 2018.
Boccardo F., Traverso M., Vizzini A., Zotti M. (2008) - Funghi d’Italia – Zanichelli, Bologna.
Bottalico A., Perrone G. (2002) – Micotossine dei macromiceti velenosi. Atti del 2° convegno Internazionale di Micotossicologia.Viterbo, 6-7 dicembre 2001. Pagine di Micologia, 17: 43-62.
Brunelli E. (2006) – Le nuove sindromi. Atti del 3° convegno Internazionale di Micotossicologia. Reggio Emilia, 6-7 Dicembre 2004. Pagine di Micologia, 25: 15-20.
Cocchi L. (2009) -Radioattività e metalli pesanti. Gli elementi chimici nei funghi superiori. In Follesa: Manuale tecnico-pratico per indagini su campioni fungini. Campioni ufficiali e non ufficiali. Intossicazione da funghi. AMB Fondazione Centro Studi Micologici. Trento.
D'antuono G., Tomasi R. (1988) – I funghi velenosi. Tossicologia micologica e terapia clinica. Edagricole, Bologna.
Denis R.B. (1995) – Mushrooms poison and panaceus. A handbook for naturalists, mycologists, and physicians. V.H. Freeman and Company. New York.
Festi F. (1985) – Funghi allucinogeni, aspetti psicofisiologici e storici. 86-ma pubblicazione dei Musei Civici di Rovereto. Manfrini R. Arti Grafiche Vallagarina, Trento.
Giacomoni L. (1999) – Une conception moderne en mycotoxicologie. Les Champignons a "toxicité variable". Atti del 1° convegno Internazionale di Micotossicologia. Roccella Jonica (RC) 4-5 dicembre 1998. Pagine di Micologia, 11: 83-86.
Heim R. (1978) – Les champignons toxiques et hallucinogenes. Boubée, Paris.
Ispano M., Strozzi M. (2006) - Allergia e intolleranza alimentare da funghi. Pagine di Micologia 25: 21-24.
Petrini O., Cocchi L., Vescovi L., Petrini L. (2009) - Chemical elements in mushroom: their potential taxonomic significance. Mycological Progress 8: 171-180.
Rascol J.P. (1999) – Toxines des champignons supérieurs et pollution. Synthèse générale des principales connaissances. Atti del 1° convegno Internazionale di Micotossicologia. Roccella Jonica (RC) 4-5 dicembre 1998. Pagine di Micologia, 11: 33-59.
Saviuc P., Cabot C., Danel V. (2005) - Allergie et champignon supérieurs. Congrés annuel de la Société Française de Toxicologie “Allergies et Toxiques”. Brest.
Sitta N., Angelini C., Balma M., Berna C., Bertocchi C., Bragalli A., Cipollone R., Corrias S., Donini M., Ginanneschi L., Gioffi D., Golzio F., Granati P., Panatta M., Tani O., Tursi A., Suriano E. (2020) – I funghi che causano intossicazioni in Italia. analisi dei dati provenienti da Centri micologici di differenti regioni e valutazioni complessive sulle intossicazioni da specie commestibili. 6° convegno Internazionale di Micotossicologia. Perugia 23-24 novembre 2018. Pagine di Micologia 41: 23-80.
Testa E. (1984) – Le intossicazioni da funghi eduli. Boll. Gr. Mic. Bres. Trento: 27 (1-2): 72-85.
Zambreri L. (1986) – I funghi nell'alimentazione. Gruppo Micologico e Protezione Flora Spontanea, Dopolavoro Ferroviario, Verona.
One a related note, several of the species in Table 1 are listed as edible or suspected not-edible. Shouldn’t inclusion in that list automatically warrant upgrading those species to not-edible or worse?
As mentioned in the text, poisoning cases can also be related to improperly mushroom collection, transport, storage, and cooking. Therefore, the species edibility was maintained for species included in Table 1 according to the classification performed by expert mycologists. We are not in the position to change this classification.
Include the sequences or primers ITS1 and ITS4-B (line 191). There is no sense in making others have to hunt for this information.
Done
In Table 2 in the PDF that I received, the accession numbers are all missing. Please make sure that the table is properly formatted.
At the time of the submission the accession numbers were not still available. Now, the sequence accession numbers were reported in the manuscript.
Line 456-457, the group that includes 47.2% can be called “a plurality”, but not “most”.
Modified
Reviewer 5 Report
The authors have generated an ITS gene dataset containing 5977 sequences to facilitate the mushroom identification with the ITS sequences mainly retrieved from GenBank and BOLD. Overall, 7270 sequences were found in databases. The authors reported that this dataset was able to identify species involved in regional poisoning incidents and support the official control activities to detect frauds and safeguard consumers.
I have some major technical concerns about the experimental design, which I believe contains severe flaws. I strongly believe that this manuscript lacks serious scientific significance. In other word, in order to identify an organism (mushroom), it is not necessary at all to build a dataset because GenBank is such a comprehensive dataset that is used internationally to identify an organism (any organisms) based on sequences. All of the proposed applications and analyses of this dataset of ITS sequences established in this manuscript are already available at GenBank with much better performance. What is the point to retrieve these ITS sequences from GenBank? How can this smaller dataset be conveniently used by public? I believe the answers to these questions are all negative based on the presentation of this manuscript. It is not explained well at all how such a small subset of ITS sequences retrieved from GenBank can be used to identify a mushroom rather than using directly GenBank with rapidly updated sequences on a daily basis. As a matter of fact, it is not necessary at all to write a paper or report for publication just to explain such a routine procedure of using GenBank to identify a mushroom (or any other organisms) based on an ITS sequence. The scientific merits are not justified and are actually not seen at all in this manuscript.
A couple of minor issues: How does public access to this proposed ITS dataset? How is the dataset updated? Based on GenBank and BOLD?
Author Response
The authors have generated an ITS gene dataset containing 5977 sequences to facilitate the mushroom identification with the ITS sequences mainly retrieved from GenBank and BOLD. Overall, 7270 sequences were found in databases. The authors reported that this dataset was able to identify species involved in regional poisoning incidents and support the official control activities to detect frauds and safeguard consumers.
I have some major technical concerns about the experimental design, which I believe contains severe flaws. I strongly believe that this manuscript lacks serious scientific significance. In other word, in order to identify an organism (mushroom), it is not necessary at all to build a dataset because GenBank is such a comprehensive dataset that is used internationally to identify an organism (any organisms) based on sequences. All of the proposed applications and analyses of this dataset of ITS sequences established in this manuscript are already available at GenBank with much better performance. What is the point to retrieve these ITS sequences from GenBank? How can this smaller dataset be conveniently used by public? I believe the answers to these questions are all negative based on the presentation of this manuscript. It is not explained well at all how such a small subset of ITS sequences retrieved from GenBank can be used to identify a mushroom rather than using directly GenBank with rapidly updated sequences on a daily basis. As a matter of fact, it is not necessary at all to write a paper or report for publication just to explain such a routine procedure of using GenBank to identify a mushroom (or any other organisms) based on an ITS sequence. The scientific merits are not justified and are actually not seen at all in this manuscript.
We regret you did not appreciate our work. In fact, as also highlighted by other authors, the need to dispose of a well-validated dataset containing accurate sequences is essential for a proper identification of fungi, since the success of the analysis greatly depends on the technical quality and the correct taxonomic identification of the deposited sequences. In addition, among the limitations of sequence-based fungal identification is strictly dependent on the capacity of correct interpretation of BLAST results, which is also in our opinion a highly underestimated problem as it is directly related to sufficient taxonomic and molecular expertise. Therefore, a preliminary analysis (filtering) aimed at assessing the reliability of the official DNA databases and the creation of curate sequences databases is fundamental to achieve the model of fungi sequence-based identification. In particular, the development of ITS dataset targeting specific groups of mushrooms could represent a valuable approach for a rapid identification via DNA barcoding.
A couple of minor issues: How does public access to this proposed ITS dataset? How is the dataset updated? Based on GenBank and BOLD?
The dataset is being developed within a project funded by the Italian Ministry of Health. As specified in Giusti, A., Ricci, E., Gasperetti, L., Galgani, M., Polidori, L., Verdigi, F., ... & Armani, A. (2021). Molecular Identification of Mushroom Species in Italy: An Ongoing Project Aimed at Reinforcing the Control Measures of an Increasingly Appreciated Sustainable Food. Sustainability, 13(1), 238 the project aims “to develop an analytical method to optimize both the management of poisoning incidents and the official control activity of nationally-marketed mushroom products. In particular, the project provides the setup of an ITS-based genetic dataset to support the identification of the wild and cultivated mushroom species in the Italian territory. The output of the project is completely pursuant to the requitement of Article 98 of the Regulation (EU) n. 2017/625 regarding the ‘Responsibilities and tasks of the European Union reference centers for the authenticity and integrity of the agri-food chain’ [47]. In fact, the EU reference centres for the authenticity and integrity of the agri-food chain are responsible for different tasks, among which is: “Where necessary, establishing and maintaining collections or databases of authenticated reference materials”. In particular, the EU reference centres collaborate with the national reference laboratories for the development and the validation of new methods. The project aims to provide technical support to the National Health Service’s expert personnel for the identification of mushroom species. Once set up and validated, the internal database will be made available to all of the national EZI network, and its access will be extended to other official laboratories operating at the international level. In addition, it can be subsequently implemented with new reference sequences of species of interest in other regions”. Therefore, the dataset is not open to the general public but only to official laboratories that can continuously update the system.
Round 2
Reviewer 2 Report
Review of the manuscript “Building of an Internal Transcribed Spacer (ITS) gene dataset to support the Italian Health Service in mushrooms identification” submitted to the journal of Foods. In the revised version, the authors tried to address the comments and many of the issues are resolved. I still believe the manuscript readability and presentation can be improved however there are no major concerns to prevent me to approve this version.
Author Response
Thank you very much for the suggestions that help us to improve the quality of the paper
Reviewer 5 Report
I have carefully reviewed the manuscript. I have no further comments on the revised manuscript.
Author Response
Thank you very much